# Risk Factors for Cytomegalovirus Infection and Its Impact on Survival after Living Donor Liver Transplantation in South Korea: A Nested Case-Control Study

**DOI:** 10.3390/pathogens12040521

**Published:** 2023-03-27

**Authors:** Seung Hyuk Yim, Mun Chae Choi, Deok-Gie Kim, Eun-Ki Min, Jae Geun Lee, Dong Jin Joo, Myoung Soo Kim

**Affiliations:** Department of Surgery, The Research Institute for Transplantation, Yonsei University College of Medicine, Seoul 03722, Republic of Korea

**Keywords:** liver transplantation, cytomegalovirus, living donor

## Abstract

Cytomegalovirus (CMV), a common pathogen, causes infectious complications and affects long-term survival after transplantation. Studies examining living donor liver transplantation (LDLT) are limited. This study analyzed the risk factors for CMV infection and its impact on the survival of LDLT patients. A nested case–control design retrospectively analyzed data from 952 patients who underwent LDLT from 2005–2021. The incidence of CMV infection for the study cohort was 15.2% at 3 months for LDLT patients managed preemptively. Patients with CMV infections were matched with those without the infection at corresponding time points (index postoperative day) in a 1:2 ratio. Graft survival was significantly lower in the CMV infection group than in the control group. CMV infection was an independent risk factor for graft survival in the matched cohort (HR 1.93, *p* = 0.012). Independent risk factors for CMV infection were female sex (HR 2.4, *p* = 0.003), pretransplant MELD (HR 1.06, *p* = 0.004), pretransplant in-hospital stay (HR 1.83, *p* = 0.030), ABO incompatibility (HR 2.10, *p* = 0.009), donor macrovesicular steatosis ≥10% (HR 2.01, *p* = 0.030), and re-operation before index POD (HR 2.51, *p* = 0.035). CMV infection is an independent survival risk factor, and its risk factors should be included in the surveillance and treatment of CMV infections after LDLT.

## 1. Introduction

CMV (cytomegalovirus) is a common pathogen that causes infectious complications and affects the long-term survival of patients after solid organ transplantation [1]. The incidence of CMV replication (CMV infection) varies from 46% to 91% according to the pretransplant CMV seropositivity of donors and recipients in the absence of prophylaxis treatment [2]. CMV infection is an independent risk factor for graft rejection in organ transplant recipients [2].

The CMV disease has related symptoms and a reported incidence of 18–29% after liver transplantation (LT) [3]. CMV disease has been identified as an independent risk factor for the survival of LT recipients in previous studies [4]. International guidelines recommend prophylaxis treatment with antiviral drugs or preemptive management for CMV with surveillance once weekly for 3–4 months in intermediate-to-high-risk groups [5]. However, information on CMV infections associated with living donor liver transplantation (LDLT) is limited [6,7,8]. In an Asian single-center study, CMV infection had an incidence rate of 13% in patients with LDLT and did not impact patient survival [7]. This study analyzed risk factors for CMV infection and their impact on the survival of patients with LDLT who were managed in a preemptive setting for CMV infections in Korea.

## 2. Materials and Methods

### 2.1. Study Material

The data of 952 patients who underwent LDLT between July 2005 and December 2021 at the Severance Hospital, South Korea were retrospectively analyzed. Patients aged <18 years (n = 85), those who died, those who underwent re-transplantation 30 days after LT (n = 36), those who underwent combined organ transplantation (n = 8), and those with incomplete data (n = 70) were excluded (Figure 1). Patient characteristics prospectively collected from the institutional LT database were retrieved for eligible patients. Additional information collected from electronic medical records included CMV antibodies before transplantation and the results of a CMV polymerase chain reaction (PCR).

### 2.2. Screening and Management of CMV Infection

CMV immunoglobulin M and CMV immunoglobulin G were tested before LDLT in recipients and donors. After transplantation, CMV prophylaxis treatment was not provided because the prophylactic use of ganciclovir or valganciclovir was not covered by national insurance in Korea. Instead, CMV was managed in a preemptive setting at our institution. CMV infection was screened using PCR at 1–4-week intervals within the first 3 months of the transplantation and at 3-month intervals until 1 year postoperatively. Fever presentation in patients with LDLT was an indication for performing CMV PCR. CMV PCR was performed, either qualitatively or quantitatively. Qualitative CMV PCR was performed with a higher frequency than quantitative CMV PCR. CMV infection was defined as more than 1000 copies of CMV in the quantitative PCR or positive results in the qualitative PCR. CMV disease presented either a CMV syndrome or tissue-invasive end-organ disease. CMV syndrome was when CMV infection was attended with at least two of the following symptoms or signs: unexplained fever of at least 2 days, constitutional symptoms such as fatigue or myalgia, leukopenia, or thrombocytopenia. Tissue-invasive CMV disease was when CMV-associated hepatitis, pneumonitis, retinitis, or gastroenteritis was confirmed by biopsy [9]. CMV infection management included administering intravenous ganciclovir or oral valganciclovir at a treatment dose tailored to renal function, according to guidelines, until PCR negativity was achieved [5].

To perform quantitative real-time PCR for CMV, DNA was extracted from whole blood using a QIAamp DSP DNA Mini Kit (Qiagen, Hilden, Germany) and QIAcube (Qiagen, Germany) in accordance with the manufacturer’s instructions. Real-time PCR for CMV-DNA was performed using the LightCycler 480 (Roche Diagnostics, Mannheim, Germany) and Bio-Core CMV Quantification real-time PCR kit (Bio-Core, Seoul, Republic of Korea). For the standardization of the results, the World Health Organization International Standard for human CMV for nucleic acid amplification techniques was used.

### 2.3. Nested Case-Control Study

Patients diagnosed with CMV infection were matched with those without the infection at specific time points (the “index postoperative day” [POD]) during follow-up. The year LDLT was performed was used as the matching variable, and the ratio of the CMV infection group to controls was 1:2. Patients assigned to the control group at specific time points were reused as potential matched controls for the next index POD of the CMV infection group, except in cases where CMV infection had occurred. If patients in the matched controls experienced CMV infection after the index POD, they were censored during the time of CMV infection during the survival analysis after the index POD.

### 2.4. Statistical Methods

Data for baseline characteristics and immunosuppressant use after LDLT were represented as a number (proportion) for categorical variables and as a median (interquartile range [IQR]), or as the mean ± standard deviation for continuous variables according to their normality. Comparisons between the CMV infection and control groups were performed using the Chi-square test, Student’s *t*-test, or Wilcoxon rank-sum test where appropriate. Risk factor analysis was performed using logistic regression analysis. To explore the impact of CMV infection on survival after LDLT, graft survival (death or re-LT) after index POD was compared between the CMV infection and control groups using the Kaplan–Meier curve with the log-rank test. In addition, the association between CMV infection and graft survival was evaluated using univariate and multivariate Cox regressions. For logistic and Cox regression, variables with *p* < 0.1 in the univariable model were included in multivariable models. All analyses were performed using the R statistical package, version 4.2.0, for macOS (http://cran.r-project.org, (accessed on 12 February 2023)), with the threshold for significance set at *p* < 0.05.

## 3. Results

### 3.1. Incidence of CMV Infection in Entire Patients

The cumulative incidence of CMV infection was 9.1%, 15.2%, 16.6%, and 16.9% at 1, 3, 6, and 12 months, respectively (Appendix A) among the 952 patients with LDLT. Among the 205 patients who experienced CMV infection, 191 (93.2%) had LDLT within the last 12 months, and 166 (80.9%) had LDLT within the last 3 months. The median time from LDLT to CMV infection was 28 (IQR 13–53) days (Appendix A).

### 3.2. Baseline Characteristics of Matched Case-Control Patients

As shown in Table 1, the median ages for patients with CMV infection and those in the control groups were 54 and 55 years, respectively (*p* = 0.144).

There was a higher proportion of female patients in the CMV infection group than in the control group (39.8% for the CMV infection group vs. 22.3% for the control group, *p* < 0.001). Body mass index (BMI), hypertension, diabetes mellitus, and cardiovascular disease had similar proportions between the groups. The CMV infection group had lower rates of viral liver disease (49.2% vs. 60.9%) and other liver diseases, such as autoimmune liver disease and toxic hepatitis (28.1% vs. 14.5%, *p* = 0.005). The incidence of hepatocellular carcinoma (41.4% vs. 64.5%, *p* < 0.001) was lower, but that of the model for end-stage liver disease (MELD) (15 [IQR 10–25] vs. 11 [IQR 8–15], *p* < 0.001) was higher in the CMV infection group. The CMV infection group had higher rates of in-hospital stay (48.4% vs. 29.4%, *p* < 0.001) and severe encephalopathy (20.3% vs. 10.5%, *p* = 0.014) before LDLT. The proportion of patients with ABO incompatibility was significantly higher in the CMV infection group than in the control groups (34.4% vs. 22.3%, *p* = 0.016). Donor characteristics were similar between the groups. The number of packed red blood cells (RBC) transfused during the operation was higher in the CMV infection group than in the control group (4 [IQR 2–8] vs. 3 [IQR 1–6], *p* = 0.013). Re-operation before index POD was marginally higher in the CMV infection group than in the control group (12.5% vs. 6.2%, *p* = 0.058).

Analysis of the pretransplant CMV antibody status in all patients in the CMV infection group showed recipient positivity (R+) regardless of donor antibody. In the control group, 253 (98.8%) patients showed recipient positivity (R+), while only three patients (1.2%) showed recipient negativity and donor positivity (R-D+). In the CMV infection group, CMV infections were diagnosed using quantitative PCR in 114 patients (89.1%) and qualitative PCR in 14 patients (10.9%).

### 3.3. Pretransplant Blood Tests and Immunosuppression

The pretransplant blood tests showed that the white blood cell counts (4.5 [IQR 2.9–6.4] 10^3^/μL vs. 3.4 [IQR 2.6–4.6] 10^3^/μL, *p* < 0.001), neutrophils (3.1 [IQR 1.8–4.7] 10^3^/μL vs. 2.1 [IQR 1.5–3.2] 10^3^/μL, *p* < 0.001), and serum glucose levels (119 [IQR 92–158] mg/dL vs. 105 [IQR 90–137] mg/dL, *p* = 0.046) were higher in the CMV infection group than in the control group (Table 2).

Tacrolimus was administered to almost all patients in both groups (99.2% vs. 98.4%, *p* = 0.874). The mean and maximum tacrolimus trough levels were similar between the groups (7.3 [IQR 5.5–8.7] ng/dL vs. 7.2 [IQR 5.7–9.0] ng/dL, *p* = 0.846 for the mean level; 13.6 [IQR 9.2–17.6] ng/dL vs. 11.9 [IQR 8.8–16.9] ng/dL, *p* = 0.333 for the maximum level). In addition, the use of immunosuppressants, including tacrolimus, mycophenolate mofetil, mTOR inhibitors, and steroids, was similar between the groups.

### 3.4. CMV Infection, CMV Disease and Graft Survival

Graft survival (death or re-LT) after index POD was significantly lower in the CMV infection group than in the control group (75.7%, 70.6%, and 66.9% at 1-, 3-, and 5-year intervals in the CMV infection group vs. 92.7%, 87.9%, and 82.9% at 1-, 3-, and 5-year intervals in the control group, respectively; *p* < 0.001, Figure 2). In the univariate and multivariate Cox regression models, CMV infection was an independent risk factor for graft survival in the matched cohort (HR 1.93, *p* = 0.012) (Appendix A). Some 11 of 128 (8.6%) patients in the CMV infection group progressed to CMV disease [seven syndromes and four tissue-invasive disease (lung, n = 2; gastrointestinal tract, n = 1; liver, n = 1)].

### 3.5. Risk Factors for CMV Infection after LDLT

In univariate and multivariate logistic regression (Table 3), the independent risk factors for CMV infection were female sex (HR 2.4, *p* = 0.003), pretransplant MELD (HR 1.06, *p* = 0.004), pretransplant in-hospital stay (HR 1.83, *p* = 0.030), ABO incompatibility (HR 2.10, *p* = 0.009), donor liver macrovesicular steatosis ≥10% (HR 2.01, *p* = 0.030), and re-operation before index POD (HR 2.51, *p* = 0.035).

## 4. Discussion

In this large single-centric LDLT population, the incidence of CMV infections within 12 months was 16.9% for patients managed in a preemptive setting. Using a nested case–control design, CMV infection was independently associated with poor graft survival in the LDLT population. The risk factors for CMV infection were female sex, MELD, an in-hospital stay before LDLT, ABO incompatibility, macrovesicular steatosis ≥10%, and re-operation. CMV infection considerably affected LDLT outcomes. Consequently, risk factors should be incorporated into the surveillance and treatment of CMV after LDLT.

Compared to deceased donor liver transplant (DDLT), LDLT recipients had lower MELD and fewer comorbidities at the time of liver transplantation. Therefore, the risk of infection with LDLT is generally lower than that of DDLT [10,11]. In the preemptive setting, the incidence of CMV infections in DDLT is 40–54% within 3 months [2,12]. This study’s incidence of CMV infections in LDLT was lower than that in DDLT, at 15.2% within 3 months. The incidence identified at 3 weeks post-LDLT (13%) in the preemptive setting was similar to previous studies of LDLT patients [7]. Although the risk of CMV infection in LDLT is lower than that in DDLT, CMV infections were identified as independent risk factors for long-term graft outcomes in this study. Therefore, this study showed that management of CMV infection after transplantation is indicated in patients with LDLT.

According to international consensus guidelines, using a preemptive approach with once-weekly CMV surveillance for 3 to 4 months and universal prophylaxis treatment for CMV infections are effective and comparable methods for reducing the incidence of CMV disease and preventing graft failure in D+/R− or R+ liver transplant recipients [5]. Even so, early CMV infections are common with the preemptive approach. Universal prophylaxis treatment has reduced graft rejection and the onset of opportunistic infections [13,14,15].

The cost of CMV drugs is an important factor for initiating prophylactic treatment for CMV infections in patients with LDLT. The cost of surveillance is another important factor limiting preemptive treatment. In South Korea, the cost of liver transplants is primarily covered by the national insurance scheme. However, this insurance does not cover prophylactic CMV drug use, even though surveillance is covered by insurance. Consequently, CMV infections post-LDLT are treated in most patients through a preemptive approach [4]. In countries such as South Korea, where LDLT and ABO incompatibility commonly require additional immunosuppression, it is necessary to implement appropriate CMV prevention strategies with insurance coverage.

There is a high level of pretransplant CMV seropositivity in South Korea and Asian countries. Therefore, seropositive LDLT recipients form the majority of recipients. Consequently, R-/D+ patients are few, so the risk of CMV infection by seropositive donors is low [16,17]. As such, this study identified only three nested case–control matched R+D-CMV high-risk patients; CMV infections did not occur in these patients. Therefore, in South Korea, it is important to study recipient or donor risk factors other than CMV antibody status and tailor CMV management accordingly.

Therefore, this study provided important clinical insights into the risk factors of CMV infections post-LDLT. Among the variables identified as independent risk factors for CMV infection after LDLT in this study, higher MELD and female sex were similar to the results of previous studies [18]. In addition, ABO incompatibility was identified as an independent risk factor for CMV infection in this study. The use of rituximab, plasma exchange, and a higher need for maintenance immunosuppression are frequently used in ABO-incompatible LDLT [19,20]. Song et al. reported no difference in the risk of CMV infection between ABO-compatible LDLT and ABO-incompatible LDLT [21]. However, since that study was a CMV-targeted case–control study performed at a single center, the desensitization and immunosuppression protocols were likely different, thereby accounting for the disparity in their results and those of this study. Further studies on the relationship between ABO incompatibility-specific desensitization and immunosuppression protocols and CMV infection are warranted.

Previous studies showed that mTOR inhibitors (sirolimus and everolimus) reduce the incidence of viral diseases, including CMV infections [22,23,24]. However, in this study, there was no difference in the use of mTOR inhibitors before the index date in patients with CMV infection compared to those in the control group. As such, using mTOR inhibitors was not a significant factor in the risk factor analysis for CMV infection. Since most patients in this study had an intermediate risk (R+) for CMV infections and did not receive CMV prophylaxis, caution is needed to generalize our results. Furthermore, additional studies on LDLT patients are needed.

This study had several limitations. First, this was a single-center study that was retrospectively analyzed. Second, the CMV PCR tests performed for CMV infection screening were heterogeneous. These tests were primarily quantitative PCRs, but approximately 15% were screened using qualitative PCR. Finally, CMV PCR test intervals were 1–4 weeks. Therefore, the nested case–control design may include lead time bias.

In conclusion, CMV infection was an independent risk factor for graft survival in the patients with LDLT who were managed in a preemptive setting for CMV infections. Consequently, risk factors for CMV infections should be incorporated into the surveillance and treatment of CMV infections after LDLT.

## Figures and Tables

**Figure 1 pathogens-12-00521-f001:**
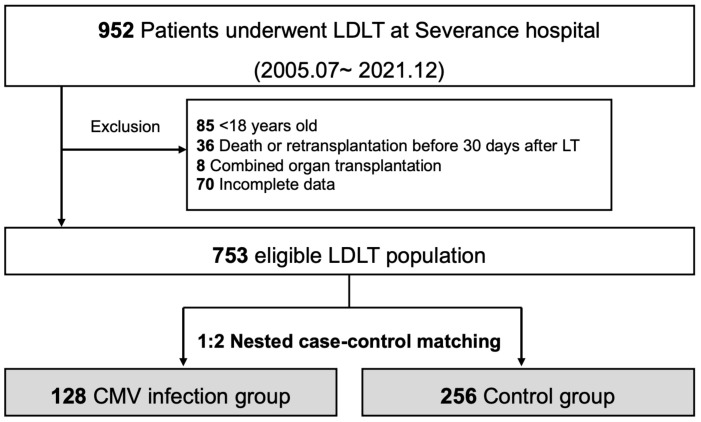
Study flow for nested case–control study.

**Figure 2 pathogens-12-00521-f002:**
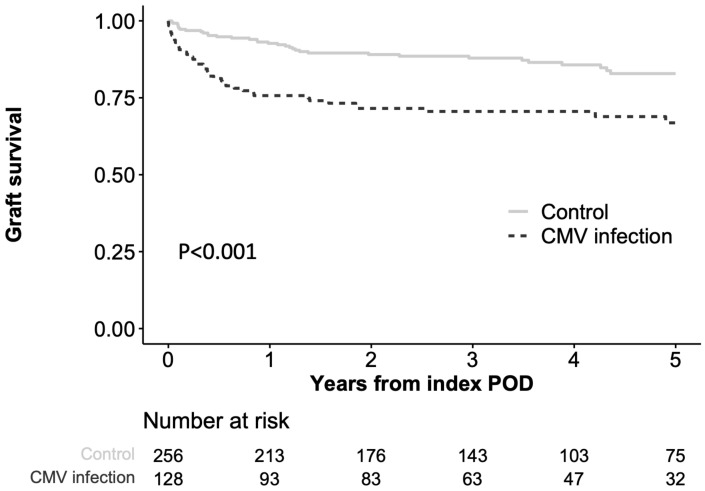
Kaplan–Meier curves for death or retransplantation after index date. The index date was set at the date of CMV viremia identification in the CMV viremia group and the corresponding date in the control group.

**Table 1 pathogens-12-00521-t001:** Baseline characteristics.

Variables	CMV Infection (n = 128)	Control (n = 256)	*p*
Age	54 (49–61)	55 (51–61)	0.144
Sex, female	51 (39.8)	57 (22.3)	<0.001
BMI, kg/m^2^	23.6 (21.5–25.4)	23.7 (22.2–25.9)	0.126
Year of LT			1.000
2012–2015	33 (25.8)	66 (25.8)	
2016–2018	70 (54.7)	140 (54.7)	
2019–2021	25 (19.5)	50 (19.5)	
Hypertension	31 (24.2)	51 (19.9)	0.403
Diabetes mellitus	42 (32.8)	82 (32.0)	0.969
Cardiovascular disease	11 (8.6)	14 (5.5)	0.342
Underlying liver disease			0.005
Viral	63 (49.2)	156 (60.9)	
Alcoholic	29 (22.7)	63 (24.6)	
Others	36 (28.1)	37 (14.5)	
HCC	53 (41.4)	165 (64.5)	<0.001
Pretransplant MELD	15 (10–25)	11 (8–15.0)	<0.001
Pretransplant stay			<0.001
Out-patient day	66 (51.6)	180 (70.6)	
In-hospital	62 (48.4)	76 (29.4)	
Refractory ascites	10 (7.8)	25 (9.8)	0.661
Severe encephalopathy	26 (20.3)	27 (10.5)	0.014
Re-transplantation	1 (0.8)	0 (0.0)	0.723
ABO incompatibility	44 (34.4)	57 (22.3)	0.016
Donor age	33 (26–43)	32 (25–41)	0.476
Donor sex, female	43 (33.6)	96 (37.5)	0.523
Donor BMI	23.0 ± 2.6	22.9 ± 2.4	0.964
Macrovesicular steatosis (≥10)	30 (23.4)	39 (15.2)	0.067
Operation time, min	646 (565–730)	652 (543–720)	0.885
Transfusion of RBC, pack	4 (2–8)	3 (1–6)	0.013
Re-operation before index POD	16 (12.5)	16 (6.2)	0.058
CMV antibody status			0.539
R+	128 (100)	253 (98.8)	
R-D+	0	3 (1.2)	
Acute cellular rejection	34 (26.6)	67 (26.2)	1.000

Data are presented as numbers (percentages), mean ± SD, or median (IQR). BMI, body mass index; CMV, cytomegalovirus; CTP, Child–Turcotte–Pugh; HCC, hepatocellular carcinoma; LT, liver transplantation; MELD, model for end-stage liver disease; RBC, red blood cell; SD, standard deviation.

**Table 2 pathogens-12-00521-t002:** Pretransplant blood tests and use of immunosuppressants.

Variables	CMV Infection (n = 128)	Control (n = 256)	*p*
White blood cell, 10^3^/μL	4.5 (2.9–6.4)	3.4 (2.6–4.6)	<0.001
Neutrophil, 10^3^/μL	3.1 (1.8–4.7)	2.1 (1.5–3.2)	<0.001
Lymphocyte, 10^3^/μL	0.6 (0.4–1.0)	0.7 (0.5–1.0)	0.077
Hemoglobin, g/dL	10.6 (8.9–12.0)	10.9 (9.2–12.5)	0.116
Platelet, 10^3^/μL	69 (51–108)	72 (52–108)	0.825
Albumin, g/dL	3.2 ± 0.5	3.2 ± 0.6	0.449
Glucose, mg/dL	119 (92–158)	105 (90–137)	0.046
Creatinine, mg/dL	0.7 (0.6–1.1)	0.7 (0.6–0.9)	0.481
Use of immunosuppressants ^a^			
TAC	127 (99.2)	252 (98.4)	0.874
Mycophenolate mofetil	82 (64.1)	165 (64.5)	1.000
mTOR inhibitor	10 (7.8)	25 (9.8)	0.661
Steroid	122 (95.3)	244 (95.3)	1.000
Mean TAC trough level, ng/dL ^b^	7.3 (5.5–8.7)	7.2 (5.7–9.0)	0.846
Maximum TAC trough level, ng/dL ^b^	13.6 (9.2–17.6)	11.9 (8.8–16.9)	0.333

Data are presented as numbers (percentages), mean ± SD, or median (IQR). ^a^: Values were acquired from the LT to index POD in each patient. ^b^: Use of immunosuppressants was defined as a prescription that was >50% of the post-transplant days before the index POD. CMV, cytomegalovirus; mTOR, mammalian target of rapamycin; TAC, tacrolimus.

**Table 3 pathogens-12-00521-t003:** Risk factor analyses for CMV infection.

	Univariable ^a^		Multivariable ^b^	
Variables	OR (95 CI)	*p*	OR (95 CI)	*p*
Age	0.98 (0.95–1.00)	0.084	0.98 (0.95, 1.01)	0.175
Sex, female	2.31 (1.46–3.67)	0.001	2.40 (1.35, 4.28)	0.003
Pretransplant MELD	1.09 (1.06–1.12)	0.001	1.06 (1.02, 1.11)	0.004
Pretransplant stay (vs. Out-patient day)				
In-hospital	2.15 (1.38–3.36)	0.001	1.83 (1.06, 3.17)	0.030
Severe encephalopathy	2.16 (1.20–3.90)	0.010	1.06 (0.47, 2.33)	0.882
ABO incompatibility	1.83 (1.14–2.92)	0.012	2.10 (1.21, 3.67)	0.009
Macrovesicular steatosis, ≥10%	1.70 (1.00–2.90)	0.050	2.01 (1.07, 3.77)	0.030
Transfusion of RBC, pack	1.03 (1.00–1.07)	0.043	1.01 (0.97, 1.05)	0.672
Re-operation before index POD	2.14 (1.03–4.47)	0.040	2.51 (1.06, 5.94)	0.035
Creatinine, mg/dL	1.71 (1.16–2.78)	0.017	1.45 (0.94, 2.51)	0.141
Glucose, mg/dL	1.00 (1.00–1.01)	0.077	1.00 (1.00, 1.01)	0.130
White blood cell, 10^3^/μL	1.18 (1.09–1.29)	0.001	0.95 (0.72, 1.26)	0.737
Neutrophil, 10^3^/μL	1.24 (1.13–1.38)	0.001	1.15 (0.84, 1.59)	0.384

^a^: Full results are provided in Appendix A. ^b^: The model was established by the enter method; covariates of which *p* < 0.1 were entered into the univariable models. BMI, body mass index; HCC, hepatocellular carcinoma; LT, liver transplantation; MELD, model for end-stage liver disease; RBC, red blood cell.

## Data Availability

The data presented in this study are available on request from the corresponding author.

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
