# Peer review of "Risk Factors for Cytomegalovirus Infection and Its Impact on Survival after Living Donor Liver Transplantation in South Korea: A Nested Case-Control Study"

_pathogens, 2023, doi:10.3390/pathogens12040521_

Round 1

Reviewer 1 Report

The subject was the patients who underwent living donor liver transplantation. The cytomegalovirus (CMV) DNA was measured regularly postoperatively for a year. The risk factor for CMV infection was examined. The authors found that female, pretransplant MELD score, pretransplant hospital stay duration, ABO compatibility and the macro-vesicular steatosis of the graft and re -operation to be risk factors for CMV infection.

I have some comments.

1.         How about the episode of acute cellular rejection as a potential risk factor for cytomegalovirus infection?

2.         The authors concluded that the CMV infection risk factors should be included in the surveillance and treatment of CMV. How will they modify the surveillance and treatment of CMV?

Author Response

  1. How about the episode of acute cellular rejection as a potential risk factor for cytomegalovirus infection?

-> As you commented, we conducted further analysis of acute cellular rejection (AR).

The AR incidence was 26.6% in the CMV infection group and 26.2% in the control group. (revised Table 1) In addition, in the univariate risk factor analysis for CMV infection, the OR of AR was 1.02 (95% CI 0.63-1.64), and the P-value was 0.935, which was not significant (revised Table S2).

  1. The authors concluded that the CMV infection risk factors should be included in the surveillance and treatment of CMV. How will they modify the surveillance and treatment of CMV?

-> Because prophylasis is difficult in South Korea due to insurance problems, it is important to carefully monitor patients with risk factors through surveilance.

Since this study is a retrospective study, there is a limit to suggesting a surveillance period other than weekly surveillance for up to 3-4 months according to the existing international consensus guidelines.

Author Response

1. Lines 251-252, since insurance doesn’t cover prophylactic use of antivirals for CMV, CMV infections should not be effectively prevented. However, the authors say CMV is prevented. Is this a typing error? If not, please explain more.

-> It is an incorrect meaning delivery caused by a typing error. We modified "CMV infections are prevented in most patients through a preemptive approach" to "CMV infections are treated in most patients through a preemptive approach".

Reviewer 3 Report

The authors present a well designed retrospective single center cohort study examining the risk factors for and outcomes of CMV infection after living donor liver transplantation. Overall the study is sound and well presented. Results are in line with previously published studies, but this study adds to the literature specifically within the domain of living donor transplantation. 

I have only two minor suggestions for improvement:

1. More details are needed on the CMV PCR assays, especially platforms and limits of detection. 

2. The p value for female sex in the univariate column in Table 3 is missing.

Author Response

  1. More details are needed on the CMV PCR assays, especially platforms and limits of detection.

-> We added the following information about the platforms and limits of detection of the CMV PCR assay to the "Method" section in revised manuscript.

To perform quantitative real-time PCR for CMV, DNA was extracted from whole blood using a QIAamp DSP DNA Mini Kit (Qiagen, Germany) and QIAcube (Qiagen, Germany) in accordance with the manufacturer’s instructions. Real-time PCR for CMV-DNA was performed using the LightCycler 480 (Roche Diagnostics, Germany) and Bio-Core CMV Quantification real-time PCR kit (Bio-Core, Korea). For the standardization of the results, the World Health Organization International Standard for human CMV for nucleic acid amplification techniques was used.

2. The p value for female sex in the univariate column in Table 3 is missing.

 -> We added about the missing data you mentioned in Table 3. (P=0.001)

Round 2

Reviewer 1 Report

The subject was the patients who underwent living donor liver transplantation. The cytomegalovirus (CMV) DNA was measured regularly postoperatively for a year. The risk factor for CMV infection was examined. The authors found that female, pretransplant MELD score, pretransplant hospital stay duration, ABO compatibility and the macro-vesicular steatosis of the graft and re -operation to be risk factors for CMV infection.

I have some comments.

1.         (L230) It is described that “CMV infection considerably affected LDLT outcomes.” Isn’t there possibility that the patients complicated after LDLT had CMV infection with high rate?

2.         Are there some data on the CMV diseases?

Author Response

  1. (L230) It is described that “CMV infection considerably affected LDLT outcomes.” Isn’t there possibility that the patients complicated after LDLT had CMV infection with high rate?

-> As shown in Table S1, although the multivariable Cox regression for graft survival was adjusted with complication related variables such as operation time, RBC transfusion, and reoperation, CMV infection was an independent risk factor for graft survival. Also, we excluded patients with early graft failure, such as mortality within 30 days or re-transplantation, from this nested case control study.

  1. Are there some data on the CMV diseases?

We added the following information about the CMV disease to the "Result 3.4" section in manuscript.

  • 11 of 128 (8.6%) patients in the CMV infection group progressed to CMV disease [7 syndromes and four tissue-invasive disease (lung, n=2; gastrointestinal tract, n=1; liver, n=1)].